# Constitutive Activation and Inactivation of Mutations Inducing Cell Surface Loss of Receptor and Impairing of Signal Transduction of Agonist-Stimulated Eel Follicle-Stimulating Hormone Receptor

**DOI:** 10.3390/ijms21197075

**Published:** 2020-09-25

**Authors:** Munkhzaya Byambaragchaa, Jeong-Soo Kim, Hong-Kyu Park, Dae-Jung Kim, Sun-Mee Hong, Myung-Hwa Kang, Kwan-Sik Min

**Affiliations:** 1Major in Animal Biotechnology, Graduate School of Future Convergence Technology, School of Animal Life Convergence Science, Institute of Genetic Engineering, Hankyong National University, Ansung 17579, Korea; monkhzaya_b@yahoo.com (M.B.); skyjs900213@naver.com (J.-S.K.); diverace@hanmail.net (H.-K.P.); 2Institute of Genetic Engineering, Hankyong National University, Ansung 17579, Korea; 3Jeju Fisheries Research Institute, National Institute of Fisheries Science (NIFS), Jeju 63610, Korea; djkim4128@korea.kr; 4Department of Technology Development, Marine Industry Research Institute for Eastrim (MIRE), Uljin 36315, Korea; hongsunmee@mire.re.kr; 5Department of Food Science and Nutrition, Faculty, Hoseo University, Asan 31499, Korea; mhkang@hoseo.ac.kr

**Keywords:** eelFSHR, constitutively activating mutation, inactivating mutation, cAMP response, signal transduction, cell surface loss of receptor

## Abstract

In the present study, we investigated the signal transduction of mutants of the eel follicle-stimulating hormone receptor (eelFSHR). Specifically, we examined the constitutively activating mutant D540G in the third intracellular loop, and four inactivating mutants (A193V, N195I, R546C, and A548V). To directly assess functional effects, we conducted site-directed mutagenesis to generate mutant receptors. We measured cyclic adenosine monophosphate (cAMP) accumulation via homogeneous time-resolved fluorescence assays in Chinese hamster ovary (CHO-K1) cells and investigated cell surface receptor loss using an enzyme-linked immunosorbent assay in human embryonic kidney (HEK) 293 cells. The cells expressing eelFSHR-D540G exhibited a 23-fold increase in the basal cAMP response without agonist treatment. The cells expressing A193V, N195I, and A548V mutants had completely impaired signal transduction, whereas those expressing the R546C mutant exhibited little increase in cAMP responsiveness and a small increase in signal transduction. Cell surface receptor loss in the cells expressing inactivating mutants A193V, R546C, and A548V was clearly slower than in the cell expressing the wild-type eelFSHR. However, cell surface receptor loss in the cells expressing inactivating mutant N195I decreased in a similar manner to that of the cells expressing the wild-type eelFSHR or the activating mutant D540G, despite the completely impaired cAMP response. These results provide important information regarding the structure–function relationships of G protein-coupled receptors during signal transduction.

## 1. Introduction

Follicle-stimulating hormone (FSH) is a unique glycoprotein that is required for pubertal development in fish [1] and mammals [2]. The genes that encode G protein-coupled receptors (GPCRs) constitute one of the largest gene families and are critical mediators of cellular signaling [3]. FSH receptor (FSHR) is part of a subgroup of glycoprotein hormone receptors within the GPCR family and occupies a large extracellular domain [4,5,6].

Although the signal transduction of glycoprotein hormone receptors has been reported [7,8,9,10,11], researchers have yet to confirm the signal transduction of eel glycoprotein hormone receptors. A growing number of naturally occurring mutations in the FSHR gene are also associated with reproductive disorders in mammals [12]. Several groups have carried out studies using FSH beta subunit (FSHβ) and FSHR knock-out mice to elucidate the function of the receptor in the reproductive organs [5,13,14]. The FSHβ knock-out female mice were sterile, had low levels of FSH, and did not experience estrous [5,14]. Similar characteristics were detected in the FSHR knock-out female mice [13].

In humans, the mutation Asp567Gly (i.e., D567G) constitutively activates human FSHR (hFSHR) and affects patients with megalotestes [2,15]. Such patients express the D567G mutation (equivalent to D540G in eelFSHR), which induces the highest basal level of cyclic adenosine monophosphate (cAMP) production in transfected COS-7 cells [15,16]. In common with thyroid stimulating hormone receptor (TSHR) (codon 619) and luteinizing hormone receptor (LHR) (codon 564), D to g transition sties cause the constitutive activation of these receptors [6,17,18]. The constitutive activation of FSHR corresponding to the substitution of amino acid Ile for Thr (I545T) in the fifth transmembrane helix also markedly increases the dose-dependent cAMP concentration in COS-7-cells [19].

The mutation is localized in a crucial region of highly conserved amino acids in all glycoprotein hormone receptors, and within the FSHRs of different species [6,18,20]. The inactivating mutations of FSHR constitute a novel pathogenic variant [21,22,23,24,25]. The cAMP responsiveness of these receptors is completely impaired by treatment with high concentrations of agonist [26,27,28]. The inactivating mutation Ala189Val which is equivalent to A193V in eelFSHR causes hereditary hypergonadotropic ovarian failure in females [20] and suppresses spermatogenesis in men [12]. A heterozygous mutation in the glycosylation site at position 191 (Asn191Ile, which is equivalent to N195I in eelFSHR) in a healthy, fertile woman completely abolished the cAMP response to FSH [16]. The inactivating mutation at position 575 (A575V, which is equivalent to A548V in eelFSHR) is conserved in three glycoprotein hormone receptors and is defective in terms of cell surface trafficking. It has been identified in a woman with hypergonadotropic hypogonadism [27].

Examination of the first amino acids in extracellular loop 1 showed that substitutions at Asp405, Thr408, and Lys409 abolished cAMP synthesis [29], and the inactivating FSHR mutations Asp224Val and Leu601Val led to impaired cAMP production in a heterozygous patient with premature ovarian failure [30,31]. Examination of the R634H mutation in the cytoplasmic tail of FSHR in a non-pregnant female revealed markedly reduced cell surface expression [28]. Furthermore, partial functional impairment and altered signal transduction of the receptor were found in a patient carrying genetic mutations Ile160Thr and Arg573Cys (equivalent to R546C in eelFSHR), located in the extracellular domain and intracellular loop 3, respectively [32]. The intracellular loop 3 of FSHR also plays an important role in cAMP production and interaction with β-arrestin to mediate internalization [31].

In recent years, GPCR signal transduction has been studied in detail with respect to cell surface receptor loss, constitutive internalization, constitutive endocytosis, recycling, and β-arrestin-dependent internalization [3,33,34,35,36]. Many studies have elucidated several features of the post-endocytotic trafficking of FSHR [7,37]. The phosphorylation site mutants of the C-terminal tail of rat FSHR (rFSHR) demonstrated similar binding affinity to that of the wild-type FSHR, but impaired internalization of the FSH–FSHR complex [38,39]. The newly created rFSHR mutations, rFSHR-1L, rFSHR-3L, and FSHR-(3L + CT) exhibited hFSH-induced internalization following phosphorylation [40]. Research has elucidated several features of the signal transduction of eelFSHR [1,41,42] and eelLHR [43] induced by diverse agonist treatments, but little is known about signal transduction for activation and inactivation in fish FSHR.

The present study aimed to delineate the mechanism of cell surface receptor loss by studying one constitutively activating mutation (D540G) and four inactivating mutations (A193V, N195I, R546C, and A548V) in the highly conserved residues of FSHR. We also aimed to determine how the activating/inactivating mutations affect signal transduction in the eelFSHR–FSH complex. Our study revealed a marked constitutive basal cAMP response in cells expressing the activating eelFSHR mutant and also demonstrated that the cell surface receptor loss due to the mutation occurred quickly.

## 2. Results

### 2.1. Preparation and Cell Surface Expression of Wild-Type eelFSHR and the Mutant Receptors

In previous studies, we analyzed the signal transduction of eelFSHR [1,42]. To determine how eelFSHR affects hormone-receptor interaction, we generated one constitutively activating mutation of eelFSHR in intracellular domain 3 (D540G), and four inactivating mutations of eelFSHR (i.e., A193V and N195I in the extracellular domain, R546C in intracellular domain III, and A548V in transmembrane VI (Figure 1)). We then determined whether the mutations had an effect on the cell surface expression of receptors.

The surface expression of eelFSHR was determined via an enzyme-linked immunosorbent assay (ELISA) in transiently transfected human embryonic kidney (HEK) 293 cells (Figure 2). Receptor expression was the same in the cells expressing the activating mutant (D540G) as in those expressing the wild-type eelFSHR. The expression level of wild-type eelFSHR was considered to be 100%, and the expression levels of N195I and A548V were 90%, whereas the expression level of the A193V mutant was approximately 85%. However, the expression level of the R546C mutant was higher (i.e., 125%). Although there were small differences among them, the mutants were typically expressed on the cell surface. We then determined the cAMP response and cell surface loss of receptors induced by agonist treatment.

### 2.2. cAMP Responsiveness Induced by Agonist in Activating and Inactivating Mutants

To examine the effect of receptor density on cAMP response, we transfected cells with-wild eelFSHR plasmid DNA using 6-well plates. The cells were then divided into 384 wells at a density of 10,000 cells/well. Cells transfected with wild-type eelFSHR DNA increased cAMP production in response to high concentrations of eelFSH agonist. The half maximal effective concentration (EC_50_) of the eelFSH-stimulated cAMP response was approximately 523 ng/mL. The basal and Rmax cAMP responses were 0.6 and 37.5 nM/10^4^ cells, respectively (Figure 3).

The D540G mutation induced constitutive activation of the eelFSHR, as induced by a 23.2-fold increase in the basal cAMP response without agonist treatment. The basal cAMP level in the D540G cells corresponded to 37% of the maximal response detected in the wild-type eelFSHR cells. The maximal cAMP response induced by eelFSH in the activating mutant with respect to the maximal response in the wild-type was approximately 0.66-fold, as shown in Table 1. Thus, the D540G cells did not respond to further stimulation by higher concentrations of agonist. To better understand the normal role of amino acid Asp540 in the eelFSHR, we investigated the effects of two other mutations—D540N and D540F (with a charged polar side chain and a nonpolar side chain, respectively)—at the same position. We did not detect any agonist response in the D540F mutation, but the D540N mutation caused a 14.5-fold stimulation of the basal cAMP response in the wild-type. The maximal response in the D540N mutant reached approximately 90% of the maximal response in the wild-type.

To directly assess the functional effects of the four inactivating mutations, we transiently expressed the receptors in Chinese hamster ovary (CHO-K1) cells. The eelFSHRs with the inactivating mutations (i.e., A193V, N195I, R546C, and A548V) were measured by quantifying cAMP accumulation in cells incubated with increasing concentrations of eelFSH. As shown in Figure 4, the basal cAMP response was not affected by the inactivating mutation. As predicted, signaling was completely impaired in three mutation receptors (A193V, N195I, and A548V). There was little increase in cAMP responsiveness in the R546C mutant, despite the high concentration of eelFSH (Figure 4). The EC_50_ value was 81%, and the maximal response was only 38% of the corresponding values in the wild-type eelFSHR (Table 2).

### 2.3. Cell Surface Receptor Loss Induced by Treatment with the eelFSH Agonist

We used an ELISA to measure the loss of eelFSHR expression from the cell surface to further explore the relationship between the cAMP and cell surface receptor loss. The cells were pre-incubated with 100, 500, and 1000 ng/mL eelFSH for 60 min. The data for the dose-dependent and time-dependent loss of eelFSHR expression are shown in Figure 5. Pre-treatment of the cells expressing wild-type FSHR led to a marked loss of cell surface receptors.

Next, we subtracted the level of cell surface expression in the cells incubated without agonist from that in the cells treated with eelFSH agonist for 30 min. The results were then expressed as a percentage of the loss of surface expression measured in the control cells, which had been pre-incubated without the eelFSH agonist (taken as 0% surface receptor loss). The cells expressing wild-type FSHR that had been treated with eelFSH agonist (1000 ng/mL) for 30 min exhibited a distinct loss (>60%) of cell surface receptor (Figure 6). We also found that the cells expressing the inactivating mutants (A193V, R546C, and A548V) expressed more slowly than the cells expressing the wild-type receptor. The cells expressing the A193V mutation did not exhibit any cell surface receptor loss following treatment with the agonist. Thus, the loss of cell surface receptors is consistent with the cAMP responsiveness induced by treatment with the eelFSH agonist. Specifically, the cell surface receptor loss in the cells expressing the N195I mutant was the same as that in the cells expressing the wild-type receptor.

Next, we characterized in more detail the time-course of cell surface receptor loss in the wild-type and mutant eelFSHRs (Figure 7). Cell surface expression in the wild-type cells gradually decreased until it reached 40% of the pre-treatment value. Cell surface expression in the cells expressing activating mutant D540G rapidly decreased to 45% in the first 5 min, then remained at between 46% and 50% for 60 min. There was no surface loss in the cells expressing the inactivating mutants (A193V and R546C) and surface loss in the cells expressing the A548V mutant was slower than in those expressing the wild-type receptor (Figure 7; Table 3). Surprisingly, cell surface loss of the N195I mutant decreased to 33% in the first 5 min, then remained at between 32% and 29% for pre-treatment times of up to 60 min. The rate of the agonist–receptor complexes formed by the constitutively activating and inactivating mutants of eelFSHR described above are presented in Table 3. These data clearly show that the three inactivating mutations (i.e., eelFSHR-A193V, -R546C, and -A548V) reduced the rate of cell surface loss of eelFSHR, whereas activating mutation eelFSHR-D540G and inactivating mutation eelFSHR-N195I enhanced the rate of cell surface loss of FSHR.

## 3. Discussion

The present study describes mutations that induce constitutive activation or impaired signal transduction in eelFSHR. Our findings confirm previous reports of mammalian FSHR mutations that caused primary ovarian insufficiency and elevated cAMP levels without agonist treatment. Thus, we prepared eelFSHR mutants containing single point mutations of five distinct amino acid residues that are highly conserved among glycoprotein hormone receptors including LHR and TSHR. These mutations stimulate basal cAMP responsiveness and/or attenuate agonist-induced activation of the receptor.

Our results showed that the D540G mutation in eelFSHR causes a marked increase in cAMP production without agonist treatment, suggesting that the mutation causes the constitutive activation of eelFSHR. Compared to the wild-type eelFSHR, the eelFSHR-D540G mutant resulted in a 23.2-fold increase in basal cAMP production in CHO-K1 cells, indicating that such mutants are constitutively activating, as previously reported in mammalian FSHRs [6,17]. Next, we investigated two mutants, D540N and D540F, at the same site. D540F did not produce any cAMP response, whereas D540N caused an increase equivalent to 14.5 times the basal cAMP response, compared to the wild-type receptor. These results are consistent with those of previous research, in which the D567N mutant in hFSHR (equivalent to D540N in eelFSHR) caused an increase in the basal cAMP activity compared to the wild-type receptor and increased the production of cAMP in response to a high concentration of hFSH [44].

In a previous study, we showed that L435R and D556Y in rLHR are constitutive activating mutants [9]. Cells expressing rLHR-L435R and rLHR-D556Y exhibited 47-fold and 25-fold increases in basal cAMP, respectively. However, rLHR-L435R did not respond to the agonist, and caused no further increase in cAMP accumulation. These results corroborate data from the present study, which indicate that eelFSHR-D540G is constitutively activating with regard to the basal cAMP response without agonist treatment. This suggests that these activating mutations are not species specific and may therefore act in the same way in other fish species.

As predicted from the results described above, the mutations investigated in the present study (eelFSHR-A193V, -N195I, and -A548V) have impaired signal transduction. R546C was the only mutant that produced an increase in the cAMP response (i.e., approximately 38% of the maximal response of the wild-type receptor) following treatment with a high concentration of the agonist. Thus, our results are consistent with previously reported signal transduction research. The mutants led to the loss of FSHR function in cells expressing them. Conformational changes in the mutated receptors could explain why the inactivating mutants did not produce cAMP responses, despite prolonged agonist stimulation.

Many GPCRs are internalized into endosomes via a clathrin-dependent pathway, then partly degraded in lysosomes or recycled to the cell membrane for prolonged agonist interaction [3,45,46]. We found that pre-incubating the activating mutant D540G with the agonist for 5 min caused slightly faster cell surface receptor loss than in cells expressing the wild-type receptor. Although basal internalization prohibits assessment, the constitutively activating D540G mutant is potentially capable of significantly increasing the basal cAMP response and could be a principal factor in the analysis of agonist-induced cell surface receptor loss and cAMP accumulation. The cell surface receptor losses in the cells expressing the three inactivating mutants—A193V, R546C, and A548V—were dramatically slower than in the cells expressing the wild-type eelFSHR. The results of these tests are consistent with those from the investigations of the cAMP response following treatment with the agonist. Our results indicate that the activating receptors (rLHR-L435R and -D556Y) were internalized at the same rate as the agonist-occupied wild-type rLHR [9]. Two inactivating mutants (rLHR-D383N and -R442H) exhibited a 1.5–5-fold increase in the half-life of internalization following treatment with the agonist [8].

In contrast, cell surface loss was faster in the cells expressing the inactivating mutant N195I than in the cells expressing the wild-type eelFSHR or eelFSHR-D540G. It is interesting to note that the cell surface loss of inactivating mutant N195I was faster than that of wild-type eelFSHR. Although the mechanism is not well-understood, we theorize that the N195I mutant is most likely routed to a lysosomal degradation pathway and not recycled back to the cell surface. Cell surface loss, down-regulation, and the trafficking of new receptors to the cell membrane could crucially affect the level of cell surface receptor expression. The turnover of intracellular FSHR in the cells expressing the activating mutant was faster than in those expressing the wild-type, suggesting that the loss of cell surface receptor expression was faster in the mutant cells than in the wild-type cells. Therefore, we suggest that the rate of internalization of the constitutively activating mutant is greater than that of the agonist-occupied wild-type eelFSHR.

Two GPCRs—hLHR and β2-adrenergic receptor (β2AR)—have been shown to undergo mediated divergent trafficking to distinct endosomal compartments. hLHR is internalized by distinct pre-early endosomes for recycling and required interactions with the LHR *C*-terminal tail, suggesting that GPCR activity can be regulated at a spatial level [47]. Research is currently under way to clarify our theory on GPCR internalization, degradation, and recycling as described in detail in [3,48]. However, further molecular studies will be required to elucidate the functional interactions of the FSHR–FSH complex in cells expressing constitutive activating and inactivating mutants.

## 4. Materials and Methods

### 4.1. Materials

The pGEM-T easy cloning vector was purchased from Promega (Madison, WI, USA). The pCORON1000 SP VSV-G tag expression vector was purchased from Amersham Biosciences (Piscataway, NU, USA). The pcDNA3 expression vector, FreeStyle™ MAX transfection reagent, Lipofectamine-3000, and FreeStyle CHO-suspension (CHO-S) cells were provided by Invitrogen (Carlsbad, CA, USA). Ham’s F-12 medium, and OptiMEM medium were purchased from Gibco BRL (Grand Island, NY, USA). CHO-K1 and HEK 293 cells were obtained from the Korean Cell Line Bank (KCLB, Seoul, Korea). A homogeneous time-resolved fluorescence (HTRF) cAMP assay kit was purchased from Cisbio (Codolet, France). Monoclonal antibodies (5A11, 11A8, and 14F5) used in the ELISA analysis and eelFSH from CHO-S cells were produced in our lab as previously reported [41]. The horseradish peroxidase (HRP) labeling of 11A8 and 14F5 monoclonal antibodies was generously performed by Medexx Inc. (Seongnam, Korea). QIAGEN Maxi plasmid kits were purchased from Qiagen Inc. (Hilden, Germany). The glass spinner flasks and disposable flasks were provided by Corning Inc. (Corning, NY, USA). All other reagents used were purchased from Sigma-Aldrich (St. Louis, MO, USA) and Wako Pure Chemicals (Osaka, Japan).

### 4.2. Site-Directed Mutagenesis and Vector Construction

Point mutations were introduced using polymerase chain reaction (PCR) strategies, and an overlap extension PCR strategy was used to create activating and inactivating mutants in eelFSHR cDNA, as previously described [43]. Two different sets of PCRs were performed, and the primer sequences used in these experiments are shown in Table 4. The full-length PCR products were cloned into a pGEM-T easy vector, and the sequence of the entire region of each mutant generated by PCR was confirmed by DNA sequencing. A schematic representation of the naturally occurring mutation sites for activating (D540G) and inactivating (A193V, N195I, R546C, and A548V) mutations in eelFSHR, is shown in Figure 1.

The mutant and wild-type eelFSHR cDNAs were subcloned into the eukaryotic expression vector pcDNA3 and pCORON1000 SP VSV-G for transfection. The plasmids were then purified, and the presence of the correct insert was confirmed by restriction enzymes. In activating the Asp540 site, we also constructed two additional mutants: D540N, which had an uncharged polar side chain, and D540F, which had a nonpolar side chain. Finally, we constructed eight receptor genes: wild-type eelFSHR, D540G, D540N, D540F, A193V, N195I, R546C, and A548V.

### 4.3. Transient Transfection

The CHO cells were transfected using the liposome transfection method, as previously described [43]. The CHO cells were cultured in growth medium (Ham’s F-12 medium containing 50 U/mL penicillin, 50 µg/mL streptomycin, 2 mM glutamine, and 10% fetal bovine serum). The HEK 293 cells were cultured in growth medium (Dulbecco’s modified Eagle’s medium containing 10 mM Hepes, 50 µg/mL gentamycin, and 10% fetal bovine serum).

The CHO cells and HEK 293 cells were grown to 80–90% confluence in 6-well plates, and the plasmid DNAs were transfected using Lipofectamine reagent. After the diluted DNA had been combined with Lipofectamine samples, the mixture was incubated for 20 min. The cells were then washed with Opti-MEM, and the DNA-Lipofectamine complex was added to each well. After 5 h, growth medium containing 20% fetal bovine serum was added to each well. The CHO cells were used for cAMP analysis 48 h after transfection. The HEK 293 cells were used to investigate surface receptor loss.

### 4.4. Production and ELISA Analysis of Recombinant eelFSH Protein (rec-eelFSH)

For rec-eelFSH production, the expression vectors were transfected into CHO-S cells using the FreeStyle™ MAX reagent transfection method as previously described [1]. One day prior to transfection, the CHO-S cells were passaged at 5 × 10^5^ cells/mL. The spinner flasks were placed on an orbital shaking platform rotating at 120–135 rpm, at 37 °C in a humidified atmosphere comprising 8% CO_2_. On the day of transfection, 260 µg of plasmid DNA and 260 µL of FreeStyle™ MAX reagent was diluted with Opti-PRO™ SFM to produce a total volume of 8 mL, which was gently mixed by inverting the tube. The mixed DNA-FreeStyle™ MAX was incubated for 10 min at room temperature (RT) to allow complexes to form, then slowly added to 200 mL of the medium containing the cells. The cell cultures were incubated at 37 °C in a humidified atmosphere comprising 8% CO_2_ on an orbital shaking platform rotating at 135 rpm. Finally, the culture media were collected on day 7 after transfection and centrifuged at 100,000× *g* for 10 min at 4 °C, to remove cell debris. The supernatants were collected and frozen at −80 °C. The samples were concentrated using either a Centricon filter or by freeze-drying, then mixed with phosphate-buffered saline (PBS). The concentration of rec-eelFSH was determined via an ELISA that had been developed previously in our laboratory [41].

The rec-eelFSH was quantified using a double-sandwich ELISA performed on plates coated with the monoclonal antibody, 5A11, which was directed against the α-subunit of eelFSH. After blocking with 1% skim milk, 100 µL of the rec-eelFSH protein was added to the wells, which were incubated for 1–2 h at 37 °C. HRP-conjugated anti-eel 11A8 antibody was then added and the plates were incubated for 1 h at RT. The wells were washed five times and incubated with 100 µL of substrate solution (tetramethylbenzidine) for 20 min at RT. The reaction was stopped by adding 50 µL of 1 M H_2_SO_4_. Absorbance at 450 nm was measured in each well using a microplate reader (Cytation 3; Biotek, Winooski, VT, USA).

### 4.5. cAMP Analysis by Homogeneous Time-Resolved Fluorescence (HTRF)

The accumulation of cAMP in the CHO-K1 cells expressing wild-type eelFSHR or the eelFSHR mutants was measured using cAMP Dynamic 2 competitive immunoassay kits (Cisbio Bioassays, Codolet, France), as described previously [1]. Cells transfected with eelFSHR-WT or the eelFSHR mutants were added to a 384-well plate (10,000 cells per well). The cells were stimulated by incubation with the rec-eelFSH for 30 min at RT. The assay was terminated by incubation with the Cisbio detection reagents, cAMP-d2 and anti cAMP-cryptate (diluted five-fold in lysis buffer, 5 µL/well), for 1 h at RT. cAMP was detected by measuring the decrease in homogeneous time-resolved fluorescence (HTRF) energy transfer (665 nm/620 nm) using an Artemis K-101 HTRF microplate reader (Kyoritsu Radio, Tokyo, Japan). This method uses an immunoassay in which the native cAMP produced by the cells competes with cAMP labeled with the dye, d2. Tracer binding is visualized using a monoclonal anti-cAMP antibody labeled with Eu3^+^ Cryptate. The specific signal-Delta F (energy transfer) is inversely proportional to the concentration of cAMP in the standard or the sample. The results were calculated from the 665 nm/620 nm ratio and are expressed as Delta F% (cAMP inhibition), according to the following equation:[Delta F% = (standard or sample ratio − mock transfection) × 100/mock transfection].(1)

The cAMP concentrations for the Delta F% values were calculated using GraphPad Prism software (GraphPad, Inc., La Jolla, CA, USA).

### 4.6. Agonist-Induced Cell Surface Loss

Loss of eelFSHR from the cell surface was assessed by an ELISA, as described previously [33,49]. The cells were plated at a density of 6 × 10^5^ cells per 60 mm dish, then split into 96-well dishes (1 × 10^4^ cells) coated with poly-d-lysine 24 h post-transfection. In the experiment to determine cell surface loss, the cells were pre-incubated with or without rec-eelFSH (1000 ng/mL) for 30 min at 37 °C. We also investigated cell surface loss of the receptors with respect to time. The cells were pre-incubated with 1000 ng/mL rec-eelFSH for the time-dependent tests (5, 15, 30, and 60 min).

The cells were fixed using 4% paraformaldehyde in Dulbecco’s PBS (DPBS) for 5 min at RT. After washing three times with DPBS, the wells were incubated with blocking solution (Tris-buffered saline with 1% bovine serum albumin) for 30 min. The primary antibody was rabbit anti-VSVG antibody, and the secondary antibody was HRP-conjugated anti-rabbit antibody (both from Abcam, Eugene, OR, USA). Following incubation, the wells were washed four times with blocking solution. We added 80 µL of DPBS and 10 µL of SuperSignal ELISA Femto Maximum substrate (Thermo Fisher Scientific, Waltham, MA, USA) to each well. Luminescence was measured using a Cytation 3 plate reader (BioTek, Winooski, VT, USA). In the time-dependent tests, the cell surface expression levels of the wild-type and mutant eelFSHRs in the non-treatment cells were considered to be 100%.

### 4.7. Data Analysis

The Multalin multiple sequence alignment tool was used for sequence analysis. GraphPad Prism 6.0 was used to analyze cAMP production and GraFit 5.0 (Erithacus Software Limited, Surrey, UK) was used to determine the cAMP EC_50_ values and for the stimulation curve analyses. Curves fitted in a single experiment were normalized to the background signal measured for the mock-transfected cells in Figure 5 and Figure 7. The data for the mock-transfected cells in the transfected cells were subtracted from the results for the cAMP levels and cell surface receptors. Each curve was drawn using data from three independent experiments. The results are expressed as the mean ± SE of three independent experiments in Figure 5 and Figure 7. Data analysis was carried out using the One-way ANOVA Tukey’s comparison tests with GraphPad Prism 6.0 software. A *p*-value of < 0.05 was taken to indicate a significant difference between groups.

## 5. Conclusions

In the present study, we demonstrated that D540G—a constitutive activating mutation of eelFSHR—causes a significant increase in basal cAMP production, and that the cell surface loss of the receptor is faster than that of the wild-type receptor. However, the results clearly showed that the three inactivating mutants—A193V, R546C, and A548V—completely impaired the signal transduction of the agonist-mediated receptor response. Although the inactivating mutant N195I impaired cAMP responsiveness, its rate of cell surface receptor loss was equal to that of the activating mutant D540G. These findings are extremely important to our understanding of FSHR function and regulation with regard to mutations of highly conserved amino acids in mammalian glycoprotein hormone receptors. Future studies on the mutations of glycoprotein hormone receptors should attempt to identify the mechanism responsible for the structure–function relationships of GPCRs.

## Figures and Tables

**Figure 1 ijms-21-07075-f001:**
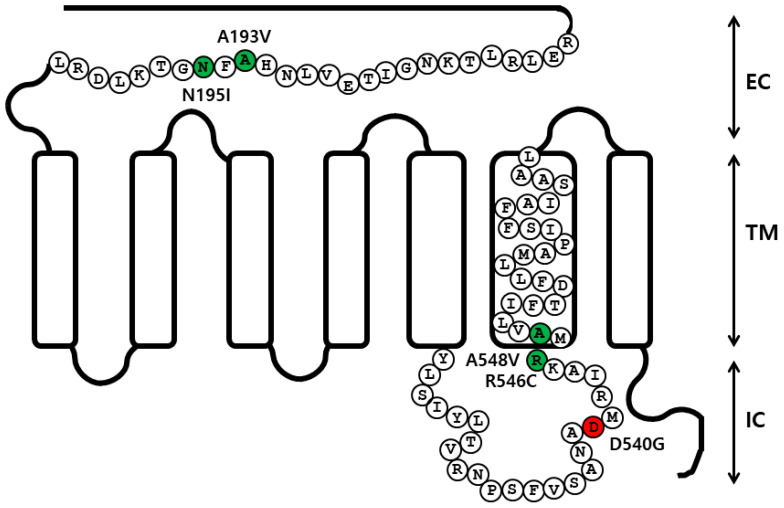
Schematic representation of the structure of eelFSHR. The locations of the constitutive activating mutation (D540G) and the four inactivating mutations (A193V, N195I, R546C, and A548V) are indicated. The red circle indicates the constitutively activating mutation and the green circles indicate inactivating mutations. EC, extracellular domain; TM, transmembrane domain; IC, intracellular domain.

**Figure 2 ijms-21-07075-f002:**
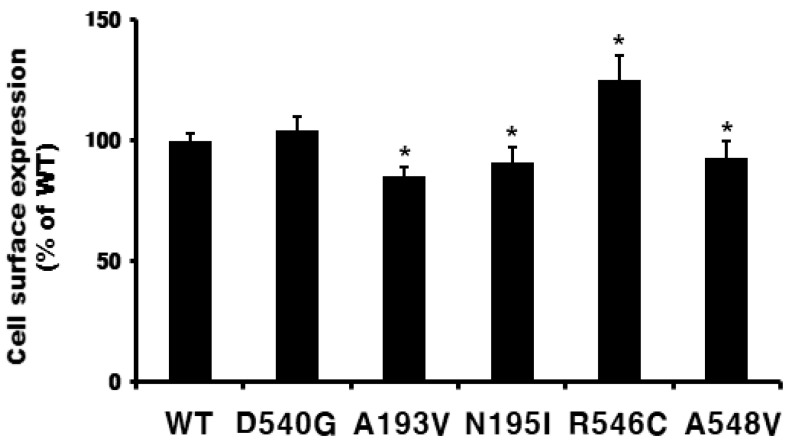
Cell surface expression of eelFSH receptors in transiently transfected HEK-293 cells. An enzyme-linked immunosorbent assay (ELISA) was used to determine the surface expression levels of the wild-type and indicated mutants of eelFSHR. Data are presented as means ± SEM of three independent experiments and were normalized to the wild-type. Cell surface expression in the wild-type was taken as 100% (see Methods and Materials). * Statistically significant differences in cell surface receptor expression (*p* < 0.05) compared to the expression of the wild-type receptor.

**Figure 3 ijms-21-07075-f003:**
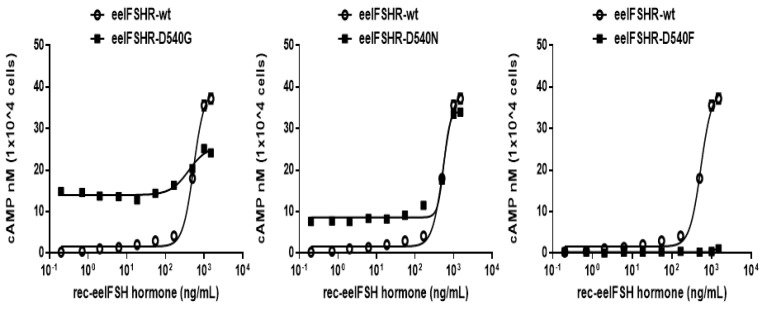
Total cAMP levels stimulated by recombinant eelFSH (rec-eelFSH) in CHO-K1 cells transfected with constitutively activating eelFSHR mutants. CHO-K1 cells transiently transfected with wild-type eelFSHR and mutants (D540G, D540N, and D540F) were stimulated with rec-eelFSH in a medium containing 0.5 mM 3-isobutyl-1-methyl xanthine for 30 min. Levels of cAMP production were determined by homogeneous time-resolved fluorescence (HTRF). The cAMP accumulation was calculated as Delta F%. The cAMP concentration was recalculated and presented using GraphPad Prism software. The mock-transfected results were subtracted from each data set. A representative data set was obtained from three independent experiments. The blank circles were the same curves of wild-type receptor.

**Figure 4 ijms-21-07075-f004:**
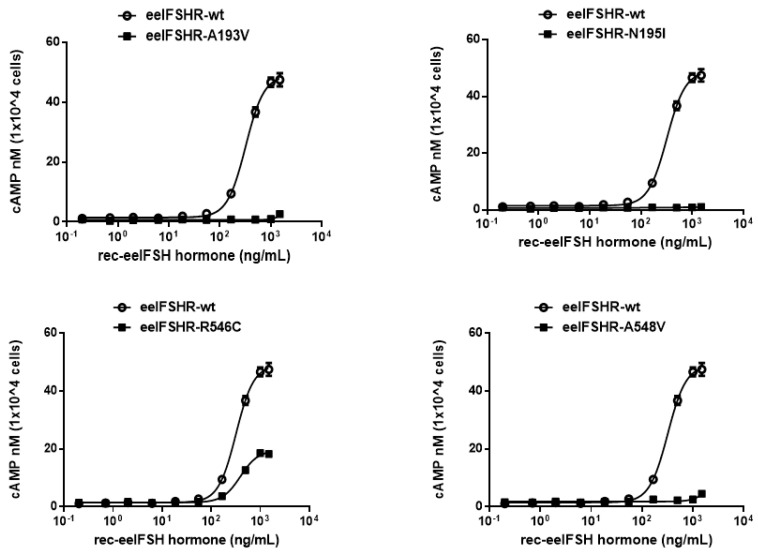
cAMP production stimulated by recombinant eelFSH (rec-eelFSH) treatment in CHO-K1 cells transfected with the inactivating eelFSHR mutants. CHO-K1 cells transiently transfected with wild-type eelFSHR and inactivating eelFSHR mutants (A193V, N195I, R546C, and A548V) were stimulated with rec-eelFSH for 30 min. Total cAMP accumulation was determined by a homogeneous time-resolved fluorescence (HTRF). The empty circles denote wild-type eelFSHR and the black circles denote the mutants. The data were subtracted from the results of the mock-transfected cells. A representative data was obtained from three independent experiments. The blank circles were the same curves of wild-type receptor.

**Figure 5 ijms-21-07075-f005:**
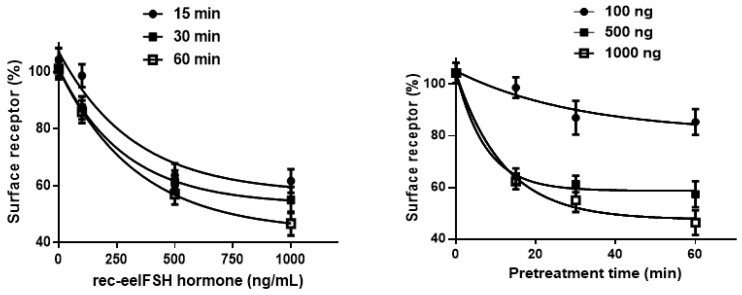
Time course and dose-dependence of the cell surface expression of wild-type eelFSHR. HEK-293 cells were transiently transfected with wild-type eelFSHR plasmid. The cells were incubated with recombinant eelFSH (rec-eelFSH) (100, 500, and 1000 ng/mL) for up to 60 min. The cell surface expression of the receptor in the non-pretreated cells was taken as 100%, as described in the Materials and Methods section. The results are expressed as the means ±SE of three independent experiments.

**Figure 6 ijms-21-07075-f006:**
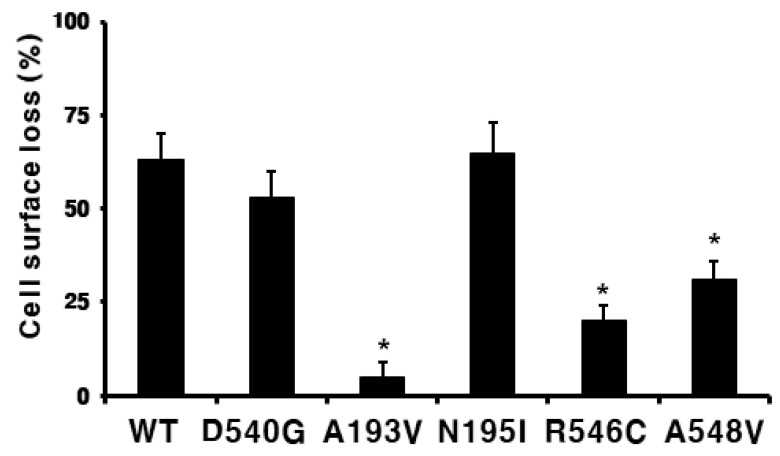
Cell surface loss of wild-type eelFSHR and activating/inactivating mutants. Each mutant plasmid was transiently transfected into the HEK-293 cells. The cells were incubated with or without 1000 ng/mL recombinant eelFSH (rec-eelFSH) for 30 min. Subsequently, the cell surface expression of the receptors was determined. The results are expressed as percentages of the cell surface loss of the receptor. Cell surface losses of the wild-type and mutant eelFSHRs were calculated by comparing levels in the presence of rec-eelFSH to levels in the absence of agonist treatment (taken as 0% cell surface loss). The results are expressed as the means ±SE of three independent experiments. * Statistically significant differences in cell surface loss of receptor (*p* < 0.05) compared to cell surface of the wild-type receptor.

**Figure 7 ijms-21-07075-f007:**
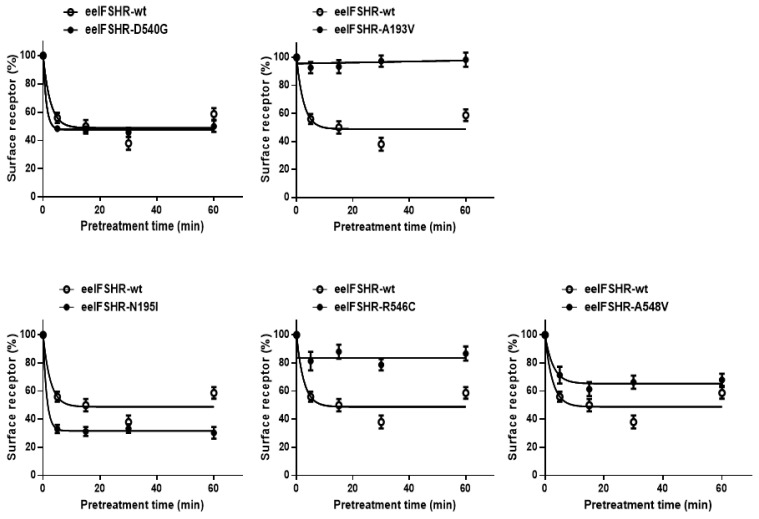
Time -dependent cell surface loss in the wild-type eelFSHR and in the activating/inactivating eelFSHR mutants. HEK-293 cells transiently expressing wild-type eelFSHR or activating/inactivating receptor were incubated with 1000 ng/mL recombinant eelFSH (rec-eelFSH) for up to 60 min. Cell surface expression in the non-pretreated groups was taken as 100% (see Materials and Methods for details). The loss of each receptor was shown using GraphPad Prism software. The results are expressed as the means ± SE of three independent experiments. In this figure, mean data were fitted to the one phase exponential decay equation. The blank circles were the same curves of wild-type receptor.

**Table 1 ijms-21-07075-t001:** Bioactivity of eelFSH receptors in cells expressing activating receptor mutants.

eelFSH Receptors	cAMP Responses		
	Basal *^a^* (nM/10^4^ Cells)	EC_50_ (ng/mL)	Rmax *^b^* (nM/10^4^ Cells)
eelFSHR-WT	0.6 ± 0.1 (1-fold)	523.8 (476.5 to 575.7)*^c^*	37.5 ± 2.6 (100%)
eelFSHR-D540G	13.9 ± 0.9 (23.2-fold)	358.8 (207.3 to 620)	24.8 ± 2.3 (66%)
eelFSHR-D540N	8.7 ± 0.7 (14.5-fold)	544.3 (399.2 to 742.3)	33.9 ± 3.7 (90%)
eelFSHR-D540F	-	-*^d^*	-

Values are the means ± SEM of three experiments. The half maximal effective concentration (EC_50_) values were determined from the concentration–response curves from in vitro bioassays. *^a^* Basal cAMP level average without agonist treatment. *^b^* Rmax average cAMP level/10^4^ cells. *^c^* Geometric mean (95% confidence limit). *^d^* Nondetectable.

**Table 2 ijms-21-07075-t002:** Bioactivity of eelFSH receptors in cells expressing inactivating receptor mutants.

eelFSH Receptors	cAMP Responses		
	Basal *^a^* (nM/10^4^ Cells)	EC_50_ (ng/mL)	Rmax *^b^* (nM/10^4^ Cells)
eelFSHR-WT	1.1 ± 0.2	314.1 (289.5 to 340.7)*^c^*	49.7 ± 5.6 (100%)
eelFSHR-A193V	0.6 ± 0.1	- *^d^*	-
eelFSHR-N195I	0.6 ± 0.1	-	-
eelFSHR-R546C	1.5 ± 0.3	385.4 (309.4 to 480.0)	18.9 ± 1.6 (38%)
eelLFSR-A548V	1.2 ± 0.1	-	-

The data are from three individual experiments. The half maximal effective concentration (EC_50_) values were determined from the concentration–response curves from in vitro bioassays. *^a^* Basal cAMP level average without agonist treatment. *^b^* Rmax average cAMP level/10^4^ cells. *^c^* Geometric mean (95% confidence limit). *^d^* Nondetectable.

**Table 3 ijms-21-07075-t003:** Rates of cell surface loss of receptors in transient transfected cell lines expressing the wild-type eelFSHR and mutants thereof.

eelFSHR Cell Lines	t_1/2_ (min)	Plateau (% of Control)
eelFSHR-WT	3.0 ± 0.8	49.9 ± 3.1
eelFSHR-D540G	1.3 ± 0.2	47.7 ± 3.8
eelFSHR-A193V	- *^a^*	-
eelFSHR-N195I	0.9 ± 0.6	31.7 ± 1.9
eelFSHR-R546C	-	82.8 ± 5.9
eelFSHR-A548V	5.7 ± 1.3	65.7 ± 4.7

Data were fitted to one phase exponential decay curves to obtain values of t1/2 and plateau (i.e., maximum reduction). The data were from three individual experiments. *^a^* Nondetectable.

**Table 4 ijms-21-07075-t004:** List of primers used to construct the eelFSHR mutants.

Primer Name		Primer Sequence
1	eelFSHR-wt Forward	5′-ATGAATTCATGTCCAATCTGCTCTTGTGGACGATG-3′ * EcoRI Site
2	eelFSHR-wt reverse	5′-CCTCGAGTTATTTAGGACCTCTGTTGAGAAT-3′ * XhoI Site
3	A193V forward	5′-GCTGAATCACGTCTTCAATGGCACC-3′
4	A193V reverse	5′-GGTGCCATTGAAGACGTGATTCAGC-3′
5	N195I forward	5′-GAATCACGCTTTCATTGGCACCAAAC-3′
6	N195I reverse	5′-GTTTTGGTGCCAATGAAAGCGTGATTC-3′
7	D540G forward	5′-GCCAATGCCGGTATGCGCATC-3′
8	D540G reverse	5′-GATGCGCATACCGGCATTGGC-3′
9	D540N forward	5′-GCCAATGCCAATATGCGCATC-3′
10	D540N reverse	5′-GATGCGCATATTGGCATTGGC-3′
11	D540F forward	5′-GCCAATGCCTTTATGCGCATC-3′
12	D540F reverse	5′-GATGCGCATAAAGGCATTGGC-3′
13	R546C forward	5′-CGCATCGCCAAGTGCATGGCCGTG-3′
14	R546C reverse	5′-CACGGCCATGCACTTGGCGATGCG-3′
15	A548V forward	5′-GCCAAGCGCATGGTCGTGCTCATC-3′
16	A548V reverse	5′-GATGAGCACGACCATGCGCTTGGC-3′

* Underlined nucleotides are the site of mutagenesis.

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
