# Peer review of "Constitutive Activation and Inactivation of Mutations Inducing Cell Surface Loss of Receptor and Impairing of Signal Transduction of Agonist-Stimulated Eel Follicle-Stimulating Hormone Receptor"

_ijms, 2020, doi:10.3390/ijms21197075_

Round 1

Reviewer 1 Report

I declared that the last version answered all my question in an acceptable way

Author Response

We don't have any resposne for Reviewer 1.

Reviewer 2 Report

Comments and suggestions for authors:

It is not clear what the error bars in Fig. 3-4 and the errors in Table 2-3 correspond to. In Figs. 5-7 the error is called SE instead of SEM, please correct.

In Fig. 2, the cell surface expressions of the mutant receptors are compared statistically with the wild type receptor. The only mention of a statistical test I could find is in line 378-388 where Student’s t-test is mentioned, but it is not clear if this is one-samle t-test. Please clarify which test was used.

In the previous submission I mentioned that the authors should show average data from all independent experiments instead of representative data in Figs. 5 and 6. According to the figure legend that now the case, but as far as I can see the data is the same. Please explain.

Please change the repeated wild type curves in Figs. 3, 4 and 7 to e.g. dashed lines or mention in the figure legends that these curves are the same.

In the previously submitted manuscript it was mentioned that the curve fits in Figs. 3 and 4 are one phase exponential decay models. This has now disappeared, but the curve fits appear to be the same. Please clarify which model was used to fit the cAMP stimulation curves and if it was the one phase exponential decay model it should be change to a sigmoidal model.

In line 374-375 it is mentioned that data in Figs. 5 and 7 are normalized to the background signal from mock-transfected cells, but another normalization is mentioned in line 363-366. Please make it clear how these data were normalized in one place.

In line 375-376 it is written that the cAMP levels from transfected cells were subtracted from mock-transfected cells, but it should be the opposite.

In line 376-378 it is written that “Each curve was drawn using data from three experiments. The results are expressed as the mean ± SEM of three independent experiments.” However, that is not the case for Figs. 3 and 4.

Author Response

Comments and suggestions for authors:

It is not clear what the error bars in Fig. 3-4 and the errors in Table 2-3 correspond to. In Figs. 5-7 the error is called SE instead of SEM, please correct.

→In Fig. 3 result was shown in the Table 1 and Fig. 4 was Table 2. We analyzed the activating and inactivating experiments, respectively. And the results by GraphPad Prism software were copied.

→We changed “SEs” to “SE” in the Fig. 5-7 Legends.

In Fig. 2, the cell surface expressions of the mutant receptors are compared statistically with the wild type receptor. The only mention of a statistical test I could find is in line 378-388 where Student’s t-test is mentioned, but it is not clear if this is one-samle t-test. Please clarify which test was used.

→In the previous 2nd Version, we wrote the One-way ANOVA Turkey’s comparison tests. But we only compared the wild type and mutant. Thus, we suggested the result of compared data with wild type receptor.

→We changed “the content of student t-test” as the One-way ANOVA Tukey’s comparison tests in the Line 375.

In the previous submission I mentioned that the authors should show average data from all independent experiments instead of representative data in Figs. 5 and 6. According to the figure legend that now the case, but as far as I can see the data is the same. Please explain.

→In the previous Version, we suggested that the SEM value was more increased than that of the data obtained the 1st submitted.

Thus, SE values in the Fig. 5 and Fig. 6 of new Version was more highly although it was not too much differences.

Please change the repeated wild type curves in Figs. 3, 4 and 7 to e.g. dashed lines or mention in the figure legends that these curves are the same.

→In the previous Version, we suggest that our GraphPad software is low version and the repeated wild type curves could not change as dashed lines. The reviewer knew the fact.

→Thus, we inserted “The blank circles were the same curves of wild type receptor.” In the Fig 3, 4 and 7 Legends.

In the previously submitted manuscript it was mentioned that the curve fits in Figs. 3 and 4 are one phase exponential decay models. This has now disappeared, but the curve fits appear to be the same. Please clarify which model was used to fit the cAMP stimulation curves and if it was the one phase exponential decay model it should be change to a sigmoidal model.

→In the 2nd Version, the curve was fit by one phase exponential decay. In the 3rd Version, we fit the Fig. 3 and Fig. 4 curves by the stimulation dose-response. Thus, the curve pattern was a little difference between 2nd Version and 3rd Version.

In line 374-375 it is mentioned that data in Figs. 5 and 7 are normalized to the background signal from mock-transfected cells, but another normalization is mentioned in line 363-366. Please make it clear how these data were normalized in one place.

→We deleted “The receptor expression level was considered to be 100% in the wild type. The levels of cell surface loss of the wild-type and mutant eelFSHRs were calculated by comparing the levels in the presence of rec-eelFSH to the levels in the absence of the agonist (considered to be 0% cell surface loss).” in the Line 363-366.

In line 375-376 it is written that the cAMP levels from transfected cells were subtracted from mock-transfected cells, but it should be the opposite.

→We changed “The data for the mock-transfected cells in the transfected cells were subtracted from the results for the cAMP levels and cell surface receptors” in the Line 371-372.

In line 376-378 it is written that “Each curve was drawn using data from three experiments. The results are expressed as the mean ± SEM of three independent experiments.” However, that is not the case for Figs. 3 and 4.

→We inserted “in Figures 5 and 7” in the Line 374.

This manuscript is a resubmission of an earlier submission. The following is a list of the peer review reports and author responses from that submission.

Round 1

Reviewer 1 Report

General comments

Figure 4 was missing from the manuscript, which makes it hard to fully review the manuscript.

Results

It is not clear what the error bars in Fig. 3-7 and the errors in Table 3 correspond to.

In Fig. 2, the cell surface expression of the mutant receptors appears to be compared with a value without error using one-way ANOVA, which is not a valid way of comparing the data. Please explain how it was done to avoid this or revise the comparison using e.g. one-sample t-test.

It is problematic that the data shown in Figs. 5 and 6 are representative experiments. Since the data is normalized in all three figures there is no reason to show data from a representative experiment instead of average data from all independent experiments.

I previously suggested to change the representation of the repeated wild type curves in Figs. 3, 4 and 7 to e.g. dashed lines. This is not difficult to do in GraphPad Prism (version 8.4.3), but if this is not possible in the software version of the authors a sentence can be inserted in the legend explaining this instead.

Curve fits have been added to Figs. 3 and 4. I cannot see Fig. 4, but according to the legend one phase exponential decay models have been used to fit the data in both Figs. 3 and 4. cAMP stimulation curves are expected to have a sigmoidal shape, so please explain why a one phase exponential decay model was used for fitting the data and how the EC50 and Rmax values were derived from fitting with this model.

The authors claim to have changed the EC50 values in Tables 1 and 2 to log(EC50), but it is simply the column headlines that have been changed, not the actual values.

Author Response

General comments

-Figure 4 was missing form the manuscript, which makes it hard to fully review the manuscript

→We rechecked all Figure and we confirmed that Fig. 4 was added to original file.

Results

-In Fig.2, the cell surface expression of the mutant receptors appears to be compared with a value without error using one-way ANOVA, which is not a valid way of comparing the data. Please explain how it was done to avoid this or revise the comparing using e.g. one-sample t-test

→We explained this point in the 4. Materials and Methods Section. 4.6 agonist-induced cell surface loss. We suggest that receptor expression level of wild type was calculated as 100%. We normalized the means data of triplicate experiment as 100. Thus, mutant’s results are compared with that of wild type receptor and displayed in the Fig. 2 as %. We also displayed that these results are done the One-way ANOVA test in the 4.7 data analysis section.

-It is problematic that the data shown in Figs. 5 and 6 are representative experiments. Since the data is normalized in all three figures there is no reason to show data from a representative experiment instead of average data from all independent experiments.

→We changed by reviewer’s comments. As 1st revision comments, we changed “triplicate out of three independent experiment” to “as a single experiment with three technical replicates” in the Fig. 5 and Fig. 6 Legends.

-I previously suggested to change the representation of the repeated wild type curves in Figs. 3, 4 and 7 to e.g. dashed lines. This is not difficult to do in GraphPad Prism (version 8.4.3), but if this is not possible in the software version of the authors a sentence can be inserted in the legend explaining this instead.

→We tried to change the wild type lines many times, but GraphPad Prism (version 6.0) do not change as dashed lines in the wild type.

-Curve fits have been added to Figs. 3 and 4. I cannot see Fig. 4. But according to the legend one phase exponential decay model have been used to fit the data in both Figs. 3 and 4. cAMP stimulation curves are expected to have a sigmoidal shape, so please explain why a one phase exponential decay model was used to fitting the data and how the EC50 and Rmax values were derived from fitting with this model.

→We changed Fig. 3 and 4 by reviewer’ comments. We fitted the Fig.3 and 4 by dose-response-stimulation method according to GraphPad Curve Fitting Guide. And then EC50 and Rmax values displayed from the beginning paper submit.

-The authors claim to have changed the EC50 values in Tables 1 and 2 to log(EC50), but is simply the column headlines that have been changed, not the actual values.

→We changed the “log(EC50)” to “EC50” by reviewer comments.

Reviewer 2 Report

the authors adequately answered our questions and we don't have any further corrections

Author Response

We don't have any response by reviewer's comments.

Round 2

Reviewer 1 Report

Figure 4 now appears in the manuscript, but the authors did not adequately answer any of my other concerns or questions.